# Impact of Molecular Weight Variations in *Dendrobium officinale* Polysaccharides on Antioxidant Activity and Anti-Obesity in *Caenorhabditis elegans*

**DOI:** 10.3390/foods13071040

**Published:** 2024-03-28

**Authors:** Xiao Pang, Heqin Wang, Chunbo Guan, Qiufeng Chen, Xinwen Cui, Xiuqing Zhang

**Affiliations:** College of Food Science and Nutritional Engineering, China Agricultural University, Beijing 100083, China; pangxiao@cau.edu.cn (X.P.); wangheqin@cau.edu.cn (H.W.); chunboguan@cau.edu.cn (C.G.); chenqiufeng@cau.edu.cn (Q.C.); cuixinwen@cau.edu.cn (X.C.)

**Keywords:** *Dendrobium officinale*, structural modification, antioxidant, obesity, *C. elegans*

## Abstract

This research investigates the impact of *Dendrobium officinale* polysaccharides (DOP) with different molecular weights on antioxidant effects, lifespan enhancement, and obesity reduction, utilizing both in vitro analyses and the *Caenorhabditis elegans* (*C. elegans*) model. Through a series of experiments—ranging from the extraction and modification of polysaccharides, Gel Permeation Chromatography (GPC), and analysis of composition to the evaluation of antioxidant capabilities, this study thoroughly examines DOP and its derivatives (DOP5, DOP15, DOP25) produced via H_2_O_2_-Fe^2+^ degradation. The results reveal a direct relationship between the molecular weight of polysaccharides and their bioactivity. Notably, DOP5, with its intermediate molecular weight, demonstrated superior antioxidant properties, significantly extended the lifespan, and improved the health of *C. elegans.* Furthermore, DOP15 appeared to regulate lipid metabolism by affecting crucial lipid metabolism genes, including fat-4, fat-5, fat-6, sbp-1, and acs-2. These findings highlight the potential application of DOP derivatives as natural antioxidants and agents against obesity, contributing to the development of functional foods and dietary supplements.

## 1. Introduction

As the global obesity epidemic escalates, over a quarter of the world’s population faces myriad health complications. According to the World Health Organization (WHO), an estimated 167 million people, encompassing adults and adolescents, are expected to suffer from health issues related to overweight and obesity by 2025 [1]. The development of obesity-related comorbidities is multifactorial, with oxidative stress playing a crucial role [2].

Current treatments for obesity range from lifestyle modifications, such as diet and exercise, to pharmacological interventions and, in severe cases, bariatric surgery. Despite the availability of these treatments, their effectiveness is often limited by factors such as accessibility, patient adherence, and long-term sustainability. Thus, there is an urgent need for innovative and effective approaches to manage and treat obesity. The quest for natural and effective bioactive compounds in food science has led to a renewed interest in polysaccharides from medicinal plants like *Dendrobium officinale*, particularly in understanding how their structural variations impact antioxidant and metabolic health benefits [3].

*Dendrobium officinale* Kimura et Migo, belonging to the *Orchidaceae* family, is recognized for its traditional medicinal and nutritional applications, having been employed in folk medicine and as a dietary supplement for millennia [4]. Notably, polysaccharides are identified as the primary active compounds, exhibiting a range of physicochemical properties and biological activities [5,6,7,8]. These activities include anticancer [9], anti-inflammatory [10], antioxidant [11], hypoglycemic [12], immunomodulatory [13], gastroprotective [14], and hepatoprotective effects [15], varying according to the extraction method used.

Prior investigations have highlighted a direct association between polysaccharides’ molecular weight (Mw) and their biological efficacy [16]. Wang’s study identified that polysaccharides with a high molecular weight (524 kDa) from mushrooms possess pronounced anti-tumor capabilities [17]. Similarly, Ren discovered significant anti-tumor activity in high molecular weight polysaccharides (463 kDa) from *Pleurotus eryngii* against HepG2 cells [18]. In addition, Xu emphasized that the molecular weight is a crucial determinant of the antioxidant effectiveness of polysaccharides. Specifically, COP2, a polysaccharide of medium molecular weight derived from camellia seed cake, demonstrated superior free radical scavenging and reducing abilities when compared to COP4, a polysaccharide with a higher molecular weight [19]. The specific relationship between the molecular weight of polysaccharides sourced from *Dendrobium officinale* and their bioactivity is still elusive. Zhang et al. investigated the apoptosis-inducing molecular weight range in HeLa cells [20], and Tao et al. observed the strongest anti-tumor effect against colorectal cancer with polysaccharides of 389.98 kDa from *Dendrobium officinale* [21]. The determination of the optimal molecular weight for *Dendrobium officinale* polysaccharides to achieve anti-obesity benefits remains an open question.

*C. elegans* offers significant advantages as a model organism in research due to its diminutive size, short lifecycle, and the conservation of genes important to human biology [22]. It is widely used to identify natural products with potential anti-aging effects and to investigate the mechanisms behind aging and stress resistance [23,24]. Additionally, the nematode’s fat metabolism pathways are closely related to those in humans, making it a powerful model for studying metabolic processes [25,26].

Building on prior lab work [27], this study intends to decompose *Dendrobium officinale* polysaccharides to examine their composition and structural differences before and after degradation. Treatments will be applied to N2 wild-type *C. elegans*, focusing on their lifespan, mobility, lipid deposition assessed by Oil Red O staining, and triglyceride measurement. The aim is to investigate how the molecular weight of *Dendrobium officinale* polysaccharides influences their anti-obesity and antioxidant effectiveness.

## 2. Materials and Methods

### 2.1. Standards and Reagents

*Dendrobium officinale* was produced in Guizhou, China. *E. coli* OP50 and *C. elegans* were maintained in the laboratory. Catalase (CAT) assay kit (A007-2-1), superoxide dismutase (SOD) assay kit (A001-1-2), total antioxidant capacity (T-AOC) assay kit (A015-1-2), and triglyceride (TG) assay kit (F001-1-1) were all purchased from the Nanjing Jiancheng Bioengineering Institute (Nanjing, China). SynScript^®^ III RT SuperMix for qPCR assay kit was procured from Qingke Biotech, Shanghai, China. All other chemical reagents used were of analytical grade.

### 2.2. Polysaccharides Extraction and Modification

The extraction and degradation of polysaccharides were performed as described earlier, with minor modifications [27]. Initially, *D. officinale* powder underwent a 200 MPa pressure treatment for 5 min with a water-to-powder ratio of 1:25. After cooling, the protein content was removed using the Sevag technique [28]. Then, ethanol was added at four times the volume of the mixture and left to precipitate overnight at 4 °C. The subsequent steps involved centrifuging at 6000 rpm for 10 min and drying the collected precipitate, producing DOP.

The modification of the H_2_O_2_-Fe^2+^ degradation method was conducted by introducing 0.3 M H_2_O_2_ and 5 μM diethylenetriamine pentaacetic acid (DTPA)- Fe^2+^ into a 2 mg/mL DOP solution, with the mixture being kept away from light for intervals of 5, 15, and 25 min. The process was followed by steps of concentration, precipitation, and drying to obtain three polysaccharide fractions, named DOP5, DOP15, and DOP25, respectively.

### 2.3. Molecular Weight Determination of Polysaccharides

A total of 0.2 g of sample was introduced into a vial with a capacity of 10 mL, and ultra-pure water was added to achieve a final volume of 10 mL. The sample was subjected to ultrasonication to ensure proper dispersion before any testing. Chromatographic separation was carried out by injecting the sample and T-series dextran into an Agilent GPC Columns PL MIXED-M (300 mm × 7.5 mm, 8 μm), using a Shimadzu Nexera UHPLC LC-30AT system for analysis. The mobile phase was ultra-pure water, maintained at a flow rate of 1 mL/min. The equation used for calculating the standard dextran curve was lgM = −1.0984t + 13.66.

### 2.4. Analysis of Polysaccharide Components

The monosaccharide composition of the sample was analyzed using a Refractive Index Differential (RID) detector and an LC-20AD system (Shimadzu, Japan). A solution was prepared by dissolving 10 mg of polysaccharide in 3 mL of 2 mol/L trifluoroacetic acid, followed by hydrolysis in an oven at 120 °C for 4 h, with the tubes sealed under nitrogen. Methanol was used to evaporate the trifluoroacetic acid post-hydrolysis.

For the derivatization step, 250 μL of the hydrolyzed sample was mixed with 500 μL of PMP-methanol (0.5 mol/L) and 250 μL of NaOH (0.6 mol/L) in a 5 mL test tube. The reaction mixture was incubated at 70 °C for 2 h in a water bath, cooled, and neutralized with 500 μL of HCl (0.3 mol/L). After mixing, the solution was diluted with 1.2 mL of deionized water, extracted with chloroform (1 mL), and centrifuged at 8000 rpm for 10 min to discard the chloroform layer, repeating the extraction twice. The final aqueous phase was filtered using a 0.22 μm filter before HPLC analysis.

### 2.5. Antioxidant Ability Detection In Vitro

#### 2.5.1. Detection of the DPPH Radical (DPPH·) Scavenging Rate

The scavenging efficacy of DOPs against DPPH radicals was investigated following a slightly modified version of the method proposed by Sun et al. [29]. To conduct this evaluation, 2 mL of 0.1 mM DPPH in ethanol was added to test tubes containing 2 mL of DOP aqueous solution at different concentrations (0.5–3 mg/mL). The test tubes were shaken and incubated in darkness at room temperature for 30 min before measuring the absorbance at 517 nm. Vitamin C (0.2–1 mg/mL) was used as a positive control. The scavenging ability against DPPH radicals was calculated using the following equation:$$\mathrm{DPPH}\text{-}\mathrm{scavenging}\;\mathrm{activity}\;(\%)\;=\;\left[1-A_{s}/A_{0}\right]\times 100$$
where A_0_ is the blank absorbance, and A_s_ is the sample absorbance.

#### 2.5.2. Detection of the Hydroxyl Radical (·OH) Scavenging Rate

To determine the scavenging efficacy against hydroxyl radicals, we utilized a modified version of a method cited in [30]. This procedure entailed combining 1 mL of a polysaccharide solution at different concentrations with 1 mL of ferrous sulfate (6 mM) and 1 mL of hydrogen peroxide (6 mM) in test tubes. The mixture was allowed to stand at room temperature for 10 min, after which 1 mL of salicylic acid was introduced. Following gentle shaking and a 30 min incubation, the mixture’s absorbance was measured at 510 nm. The hydroxyl radical scavenging rate (%) was calculated using the following equation:$$\mathrm{Scavenging}\;\mathrm{rate}\;(\%)\;=\;\left[1-\frac{A_{1}-A_{2}}{A_{0}}\right]\times 100$$

A_0_ represents the absorbance of the control group (without a sample), A_1_ is the absorbance of the experimental group (without adding water), and A_2_ is the absorbance of the blank group (without adding salicylic acid).

##### 2.5.3. Total Antioxidant Capacity (T-AOC) Assay

Total antioxidant capacity was determined spectrophotometrically using kits that were commercially available, with the results being expressed in units per milliliter (U/mL) of polysaccharides. In the procedure, 0.5 mL of Reagent 1, 0.5 mL of the polysaccharide samples (DOPs), 1 mL of the Reagent 2 working solution, and 0.25 mL of the Reagent 3 working solution were sequentially added to centrifuge tubes. The tubes were then thoroughly mixed using a vortex oscillator and incubated at 37 °C in a water bath for 30 min. Following this, 0.05 mL of Reagent 4 was added, after which the mixture was mixed again and allowed to stand for 10 min. The absorbance at 520 nm was measured, with double distilled water being used to set the spectrophotometer to zero.
$$T\text{-}\mathrm{AOC}\;(U/\mathrm{mL})=\frac{E_{1}-E_{0}}{0.01}÷30\times \frac{4.1}{0.5}$$
where the absorbance of the control group is denoted as (E_0_), and the absorbance of the sample group is denoted as (E_1_).

### 2.6. C. elegans Assays

#### 2.6.1. Culture and Synchronization of *C. elegans*

Wild-type *C. elegans* (N2) and *Escherichia coli* OP50, acquired from the Caenorhabditis Genetics Center (CGC) at the University of Minnesota, were utilized in this research. The nematodes were grown on plates with nematode growth medium (NGM) at a constant temperature of 20 °C and were provided with *Escherichia coli* OP50 as food. The synchronization of the worm population was achieved through the application of the alkaline hypochlorite technique to bleach gravid adults and collect eggs. Following a 48 h incubation period at 20 °C, the eggs developed into L4 larvae, which were subsequently selected for the experiments.

#### 2.6.2. Lifespan Assay

Following the methodology described by Wang et al. [24], the longevity of worms was assessed. NGM plates (containing FUdR) and varying concentrations of samples were prepared to achieve final DOP, DOP5, DOP15, and DOP25 concentrations of 400 mg/L, 160 mg/L, 100 mg/L, and 12.8 mg/L, respectively, in the *E. coli* OP50 culture. The control group received an M9 buffer instead. Synchronized worms were placed on these media at 20 °C, marking the commencement of the lifespan study (day 0). Each experimental setup included 30 worms per plate, with three replicates for each treatment group. The number of surviving, deceased, and censored worms was documented daily until the death of the last worm.

#### 2.6.3. Body Bend Assay

The assessment of body bend frequency was performed according to the method reported by Shi et al. [31]. Synchronized worms were placed on fresh NGM plates supplemented with OP50 *E. coli* and FUdR, then exposed to decreasing concentrations of DOP, DOP5, DOP15, and DOP25 (400 mg/L, 160 mg/L, 100 mg/L, and 12.8 mg/L, respectively) and M9 buffer on days 0, 2, and 7. Subsequently, worms were moved to NGM plates without food to count the number of body bends over a 30 s period as a measure of activity. This process was repeated for at least 20 worms per treatment group to ensure statistical reliability.

#### 2.6.4. Pharyngeal Pumping Assay

The pharyngeal pumping rate of nematodes was quantified using the protocol established by Yang et al. [32]. After being cultured to synchronize their development, the nematodes were placed on fresh NGM plates containing OP50 *E. coli* and FUdR. They were subjected to treatments with varying concentrations of DOP, DOP5, DOP15, and DOP25 (400 mg/L, 160 mg/L, 100 mg/L, and 12.8 mg/L). A control group of worms was left untreated. The treatments were conducted over the course of 0, 2, and 7 days.

#### 2.6.5. Assessment of Antioxidant Stress Ability in *C. elegans*

Following a modified protocol from Qin et al., the oxidative stress resistance of worms was assessed [33]. Initially, synchronized worms were placed on fresh NGM plates with OP50 *E. coli* and exposed to descending concentrations of DOP, DOP5, DOP15, and DOP25 (400, 160, 100, and 12.8 mg/L) along with M9 buffer for one day. Afterward, 20 worms from each treatment were moved to NGM plates with 50 mmol/L hydrogen peroxide, and mortality was recorded after 10 h. This procedure was replicated three times to ensure reliability.

#### 2.6.6. Detection of Endogenous Antioxidant Enzymes

Synchronized worms at the L4 developmental stage were cultured in different DOP concentrations for a duration of five days, and then washed with M9 buffer. The activities of antioxidant enzymes, specifically SOD and CAT, were quantified according to the assay kit’s provided protocol. This methodology was strictly adhered to as outlined in the kit’s instructions, with the entire process replicated three times for consistency.

#### 2.6.7. Analysis of Body Fat Accumulation in *C. elegans* Obesity Model

Worms were subjected to an obesity model by exposure to 10 mM glucose in NGM plates, with 10 mM glucose acting as the positive control for obesity induction. Experimental groups were treated with 10 mM glucose combined with different doses of DOP, DOP5, DOP15, andDOP25 (400 mg/L, 160 mg/L, 100 mg/L, and 12.8 mg/L), and a set of plates without glucose addition was used to establish control conditions. Moreover, orlistat, administered at 400 mg/L, functioned as the negative control, providing a comparative basis for assessing the mitigation of obesity-related outcomes.

In this investigation, the influence of DOPs on fat storage in *C. elegans* was examined through Oil Red O staining and the quantitative measurement of triglycerides (TG). 

##### Oil Red O Staining Assay

Synchronized L4 worms were grown under different dietary regimes for 36 h at 20 °C. Adapting the Oil Red O staining protocol for *C. elegans*, we dissolved 0.5% ORO in 1,2-propanediol, allowing it to stand overnight [34]. The solution was filtered using a 0.22 μm syringe filter before application. Worms were then washed off NGM plates, rinsed three times with M9 buffer, frozen at −80 °C for three freeze-thaw cycles, and washed again three times in cold M9. They were dehydrated in 1,2-propanediol for five minutes before being stained with 0.5 mL of the ORO solution at room temperature for 24 h. Following staining, the worms were washed with 85% 1,2-propanediol and PBS, resuspended in PBS, and placed on 2% agarose-coated slides for subsequent microscopic analysis.

##### TG Quantification

The study evaluated the impact of extracts on TG reduction in worms made obese through high-glucose diets. L4 stage worms were synchronized and cultured under different dietary conditions for 36 h at 20 °C. Following this, the worms were placed into centrifuge tubes and washed three times with M9 buffer [35]. TG content and BCA protein determination were performed using commercially available assay kits according to the manufacturer’s instructions (Nanjing Jiancheng Bioengineering Institute, Nanjing, China). All calculated TG concentrations were standardized by protein concentration. Each condition included three biological replicates.

#### 2.6.8. Gene Detection via q-PCR

L4 stage worms synchronized in development were treated with DOP15 and relocated daily to fresh plates. After a two-day incubation period, the worms were rinsed with M9 buffer and placed into sterile, enzyme-free EP tubes. These tubes were then centrifuged at 3500 rpm for one minute to separate the worms, discarding the supernatant afterward. The worm sediment was resuspended in M9 buffer, and this centrifugation step was repeated thrice. The extracted RNA was converted to cDNA with the SynScript^®^ III RT SuperMix for qPCR (Qingke Biosciences, Shanghai, China) and quantified by fluorescence. The PCR amplification protocol included an initial step at 95 °C for 5 min, then 40 cycles at 95 °C for 15 s, 60 °C for 20 s, and 72 °C for 20 s. The primers used are detailed in Table 1.

### 2.7. Statistical Analysis

All data were obtained from three independent replicates. Statistical analysis was conducted using SPSS Statistics version 20 (IBM, Armonk, NY, USA), employing one-way ANOVA and Tukey’s multiple comparisons to determine statistical significance. Differences with *p* ≤ 0.05 were considered statistically significant.

## 3. Results and Discussion

### 3.1. Molecular Weight Detection

There exists a profound connection between the molecular weight of polysaccharides and their biological activity. Specifically, polysaccharides with a higher molecular weight, which often exhibit poor solubility in water, can impede absorption within the body, leading to diminished bioactivity and potentially reducing their practical applications. The technique of gel permeation chromatography (GPC) is utilized to assess the molecular weight of polysaccharides. Data presented in Figure 1 reveal that the molecular weights of DOP, DOP5, DOP15, and DOP25 were identified to be 507.65 kDa, 214.97 kDa, 125.41 kDa, and 16.07 kDa, respectively.

Research conducted by Zheng et al. focused on the degradation of chitosan polysaccharides using H_2_O_2_ and vitamin C under varied reaction conditions, showing a decrease in molecular weight with prolonged reaction time [36]. Notably, despite the reduction in molecular weight, the degraded polysaccharides showed no change in monosaccharide composition, indicating the preservation of functional groups during degradation.

### 3.2. Polysaccharide Composition

Following hydrolysis with TFA and analysis using LC-MS (Figure 2), it was found that DOP and its derivatives, DOP5, DOP15, and DOP25, are heteropolysaccharides, characterized by identical monosaccharide types but differing in molar ratios. Glucose and mannose were identified as the primary monosaccharides, with glucose making up 19.19%, 54.00%, 50.83%, and 59.96%, and mannose constituting 79.47%, 39.25%, 45.04%, and 37.29% of DOP, DOP5, DOP15, and DOP25, respectively. The disparity in monosaccharide composition across samples suggests that degradation processes, possibly involving reactive radicals, transform polysaccharides into simpler forms without altering the basic structural framework of DOP [37].

Polysaccharides from *Cercis chinensis*, balsam pear, and rice bran have been identified as effective antioxidants that can prevent oxidative stress-induced aging by either reducing free radical formation or slowing oxidative reactions [38,39,40]. The antioxidant activity of different concentrations of DOPs was tested in vitro. The scavenging and total antioxidant capacity of DOP, DOP5, DOP15, and DOP25 against DPPH• and OH• radicals are summarized in Figure 3. The scavenging capacity showed a dose-dependent relationship. For DPPH• scavenging activity, DOP5 exhibited the highest scavenging rate (49.48 ± 1.33%), followed by DOP15 and DOP (40.19 ± 0.62%, 37.38 ± 0.57%, respectively). DOP-25 showed the lowest detected value (30.37 ± 1.23%). Evaluation of polysaccharides’ free radical scavenging ability against OH• is crucial. The scavenging capacity in DOP5 was also the highest (49.48 ± 1.33%), while the lowest activity was observed with DOP25 (28.37 ± 0.92%). Similarly, the highest total antioxidant capacity was found in DOP5 (3.47 ± 0.07 U/mL), followed by DOP15 and DOP (1.94 ± 0.11 U/mL, 1.36 ± 0.13 U/mL). According to the antioxidant test results, DOP5 exhibited the highest activity among all samples. These results indicate that DOP, DOP5, DOP15, and DOP25 possess antioxidant activity, with DOP5 showing the highest antioxidant activity. Such techniques for measuring the antioxidant activity and bioavailability of polysaccharides may indicate differences in the antioxidant activity of extracts. The chemical properties of polysaccharides directly affect their quenching ability. These free radicals are associated with aging, obesity, diabetes, and the breakdown of essential fatty acids. As mentioned earlier, antioxidants help prevent these diseases [41].

### 3.3. C. elegans Assays

In order to better understand the biological effects of this extract, the antioxidant and anti-obesity activity was assessed in *C. elegans*, as this nematode offers the possibility of detecting phenotypic changes.

#### 3.3.1. Effects of Polysaccharides on the Lifespan of *C. elegans*

Previous studies have indicated a correlation between a high-sugar diet and shortened lifespan in nematodes [42]. From Figure 4 and Table 2, it is evident that compared to the control group, the survival curve of high-fat nematodes induced by 10 mmol/L glucose shifted to the left, with an average lifespan of 17.38 ± 0.85 days, shortened by 9.10% (*p* < 0.05), which is consistent with the findings of Elena et al. [43]. Additionally, research suggests that the reduced lifespan of high-fat nematodes may be due to the excessive generation of reactive oxygen species during organism metabolism, leading to aggravated oxidative damage [44].

Furthermore, the effects of different molecular weight DOPs on nematode lifespan were investigated. As shown in Table 2, the average lifespan of the control group was 19.12 ± 0.55 days. With the decrease in polysaccharide molecular weight, the survival curve of nematodes initially shifted to the right and then to the left (Figure 4). After DOP treatment, the average lifespan of nematodes was 21.10 ± 0.50 days, extending by 10.32% compared to the control group. Following DOP15 intervention, the average lifespan of nematodes was 22.02 ± 0.58 days, extending by 15.19% compared to the control group. The most significant lifespan extension effect was observed in the DOP5 treatment group, with an average lifespan of 23.66 ± 1.36 days, corresponding to a 23.73% increase. As a model organism for aging, nematode lifespan serves as a critical indicator of aging. The results indicate that DOP, DOP5, DOP15, and DOP25 can extend nematode lifespan. However, DOP5 exhibits better effects in prolonging nematode lifespan and delaying aging.

#### 3.3.2. Effects of Polysaccharides on the Locomotion Capacity of *C. elegans*

Aging is associated with a gradual decline in the vitality of muscle cells, leading to diminished mobility among other aging-related phenotypes [45]. To ascertain if DOPs merely increase lifespan or also improve the quality of life and extend the period of health in aging, we assessed their impact on critical age-associated physiological functions in *C. elegans*, specifically body bending and pharyngeal pumping rates. Our analysis, performed at intervals of 0, 2, and 7 days, demonstrated that DOP, DOP5, and DOP15 significantly bolstered pharyngeal muscle contractions and increased body bending frequency, as evidenced in Figure 5a and Figure 5b, respectively. Given that decreased mobility and coordination are indicative of aging in nematodes [46], these outcomes strongly indicate that DOP5, DOP15 and DOP play a substantial role in prolonging the healthy lifespan of *C. elegans*, enhancing both mobility and vitality.

#### 3.3.3. The Ability to Resist Oxidative Stress

Figure 6 reveals that nematodes undergoing treatment with DOP compounds showed a marked increase in survival against oxidative stress induced by hydrogen peroxide. Survival rates at 10 h post-treatment for groups receiving DOP, DOP5, and DOP15 (21.99 ± 2.43%, 37.72 ± 1.35%, and 47.71 ± 3.70%, respectively) significantly exceeded (*p* < 0.001) that of the control group (13.44 ± 1.18%). Hence, it can be inferred that DOP and its derivatives potentially prolong nematode survival in adverse conditions through their antioxidative effect. This evidence supplements prior in vitro assays, establishing the antioxidative defense provided by polysaccharides of varied molecular weights. Parallel research by Chen et al. has also demonstrated the capacity of Camellia oleifera polysaccharides to lower ROS proliferation and alleviate oxidative stress [47]. To delve deeper into the antioxidative actions of DOPs, the impact of antioxidative enzymes within *C. elegans* was analyzed.

#### 3.3.4. Antioxidant Enzyme Activities in *C. elegans*

Superoxide dismutase (SOD) and catalase (CAT) play critical roles as primary defense mechanisms against oxidative stress, with their upregulation being a key target for mitigating oxidative damage linked to cardiovascular or neurodegenerative diseases. These enzymes are notably conserved across nematodes. As shown in Figure 7a,b, our findings revealed that the DOP5 treatment markedly enhanced SOD and CAT activities by 5.3-fold and 2.2-fold, respectively, in comparison to untreated controls. The DOP15 group also demonstrated an increase in enzyme activities, though to a lesser extent, with SOD and CAT activities rising by 4.8-fold and 1.5-fold, respectively. While the DOP and DOP25 treatments did elevate SOD activity, they did not significantly impact CAT activity when compared to the control group.

#### 3.3.5. Effects of Polysaccharides of Different Molecular Weights in Fat Accumulation

Oil Red O staining, a method for highlighting lipid droplets microscopically, alongside triglyceride quantification, serves as a gauge for in vivo lipid levels, where a decrease in triglycerides indicates diminished lipid synthesis, potentially lowering lipid accumulation [48]. In wild-type nematodes, heightened ROS levels have been associated with increased fat storage [49]. As depicted in Figure 8a,b, administration of DOP, DOP5, and DOP15 markedly decreases triglyceride concentrations in nematodes, in contrast to DOP25, which results in an increased triglyceride accumulation. Specifically, compared to the NC group, DOP, DOP5, and DOP15 treatments resulted in triglyceride reductions of 30.4%, 34.59%, and 50.9%, respectively, while DOP25 saw a 7.8% increase. Oil Red O staining visuals further corroborated these findings, showcasing less intense staining in nematodes treated with DOP, DOP5, and DOP15, underscoring their lipid-lowering efficacy.

#### 3.3.6. Effect of DOP15 on the Gene Expression of *C. elegans*

In exploring *C. elegans* to understand disease mechanisms and physiological behaviors, our study employed quantitative PCR to assess the influence of DOP15 on genes regulating lipid metabolism [50]. Following treatment with DOP-15, Figure 9 reveals significant downregulation in the expression of critical genes for lipid metabolism: fat-4, fat-5, fat-6, sbp-1, and acs-2. The gene sbp-1, belonging to the SREBP family, is instrumental in lipid synthesis, promoting the transcription of genes involved in fatty acid and triglyceride synthesis [42]. Acs-2, which encodes long-chain fatty acid CoA synthase, is crucial for converting fatty acids to fatty acyl-CoA, a key step in both fatty acid oxidation and synthesis. The diminished expression of sbp-1 and acs-2 points to DOP15’s role in hindering lipid synthesis, which directly impacts energy production and lipid level regulation within cells. The downregulation of desaturase genes fat-4, fat-5, and fat-6 further implies a reduction in unsaturated fatty acid synthesis. The correlation among these lipid metabolism genes suggests that DOP15 could mitigate lipid accumulation in *C. elegans*.

## 4. Discussion

The study reveals a significant relationship between the molecular weight of polysaccharides and their biological activity, with higher molecular weights leading to reduced solubility and bioactivity. Through gel permeation chromatography, varying molecular weights of polysaccharides (DOPs) were identified, influencing their antioxidant abilities. The highest antioxidant activity was observed in DOP5. Medium molecular weight polysaccharides show superior antioxidant properties, consistent with findings reported by Xu in studies of camellia seed cake [19]. Further, the study found that different molecular weight polysaccharides could extend the lifespan and enhance the health span of *C. elegans*, indicating a potential anti-aging effect. Lower molecular weight polysaccharides (DOP5, DOP15) showed better outcomes in lifespan extension, antioxidant enzyme activities, and reduced fat accumulation in *C. elegans.*

Currently, researchers are increasingly developing food and beverage formulations centered on the use of dendrobium flowers and leaves. Products such as functional jellies, tea bags, and functional beverages have been reported in studies [51,52,53]. *C. elegans*, despite its diminutive size, shares significant physiological and metabolic similarities with humans, including a high level of genetic homology [25]. Consequently, research on *C. elegans* is critical for foundational studies relevant to human health and biology [35]. As model organisms, *C. elegans* are particularly useful for exploring antioxidant and lipid metabolism due to their straightforward phenotypes and defined physiological and molecular underpinnings [2,42]. Future research should focus on exploring the biological mechanisms that enable degraded polysaccharides to enhance antioxidant activity and metabolic health. Additionally, investigating the impact of varying the degradation process on bioactivity could unveil more efficient methods for producing polysaccharide-based therapeutics.

## Figures and Tables

**Figure 1 foods-13-01040-f001:**
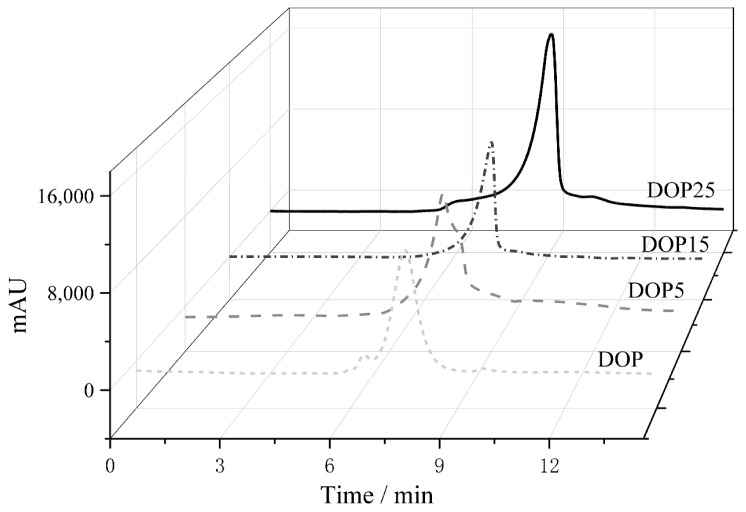
GPC spectra of DOP, DOP5, DOP15, and DOP25.

**Figure 2 foods-13-01040-f002:**
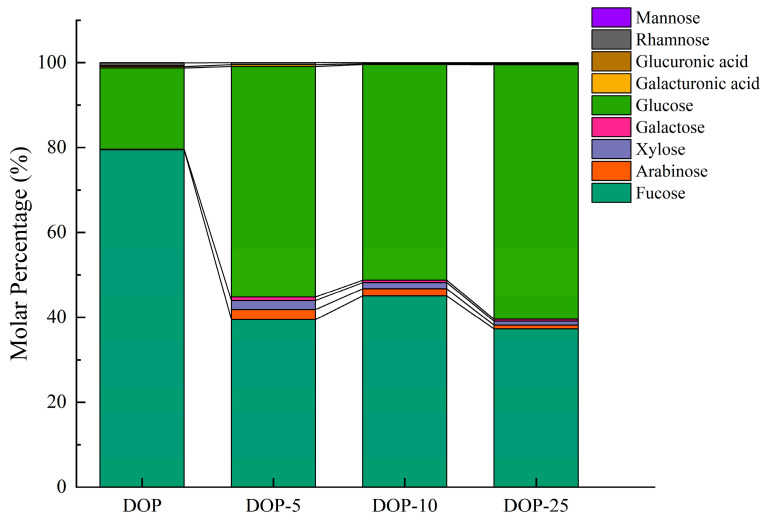
Composition of DOP, DOP5, DOP15, and DOP25.

**Figure 3 foods-13-01040-f003:**
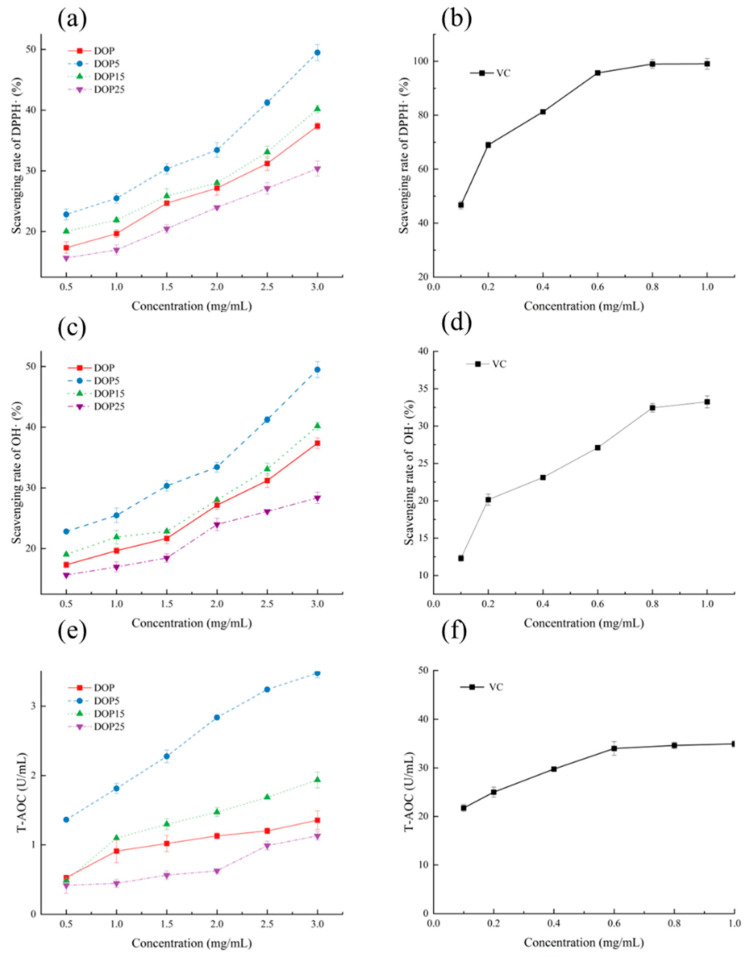
Antioxidant activity: DPPH scavenging rate of (**a**) DOP, DOP5, DOP15, and DOP25; (**b**) Vc; OH• scavenging rate of (**c**) DOP, DOP5, DOP15, and DOP25; (**d**) Vc total antioxidant capacity of (**e**) DOP, DOP5, DOP15, and DOP25; (**f**) Vc. Data are presented as mean ± SD (*n* = 3).

**Figure 4 foods-13-01040-f004:**
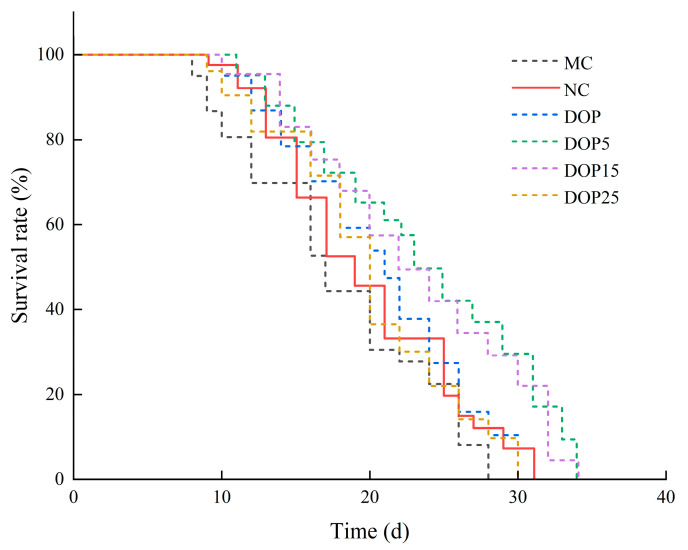
Effect of DOPs on the lifespan of *C. elegans*.

**Figure 5 foods-13-01040-f005:**
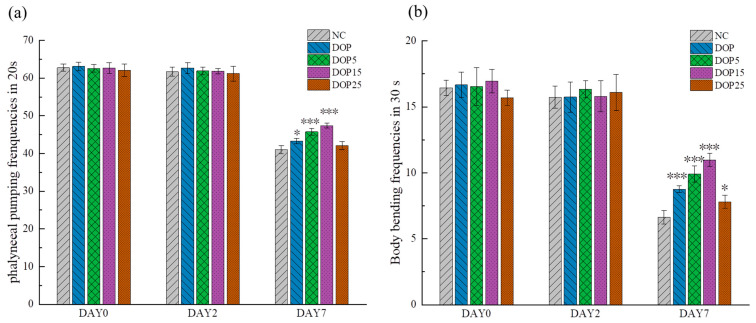
(**a**) Effect of D2-G1S-1 and D2-G1S-2 on pharyngeal pumping. (**b**) Effect of D2-G1S-1 and D2-G1S-2 on body bending frequency. Results were represented as mean ± SEM. Differences compared to the control group were considered significant at *p* < 0.05 (*) or *p* < 0.001 (***).

**Figure 6 foods-13-01040-f006:**
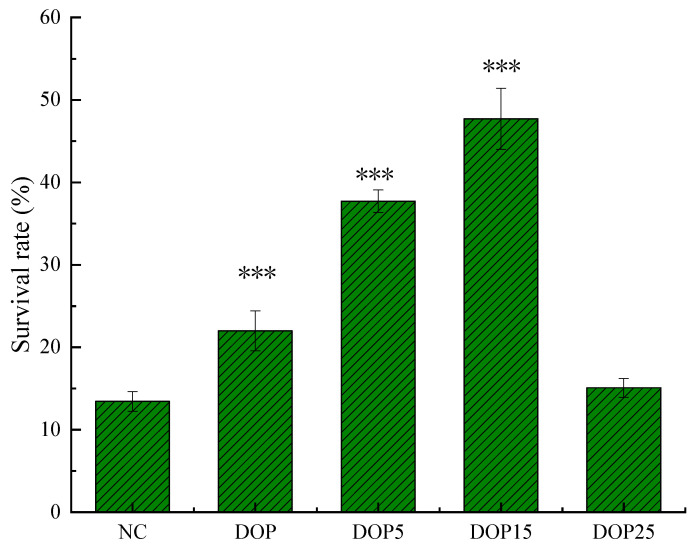
Survival rate of the *C. elegans* N2 (L4 stage) treated with 50 mmol/L H_2_O_2_ on NGM plates with or without DOPs. Results were represented as mean ± SEM. Differences compared to the control group were considered significant at *p* < 0.001 (***).

**Figure 7 foods-13-01040-f007:**
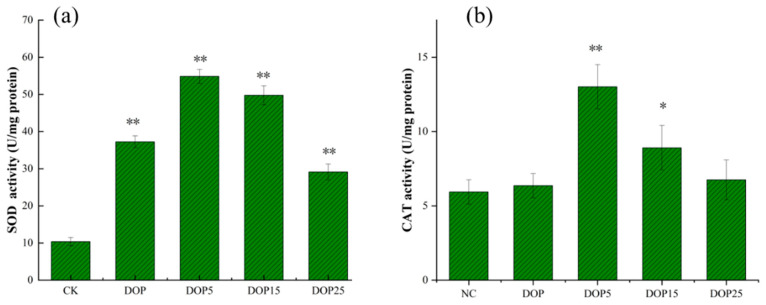
Effect of DOPs on SOD (**a**) and CAT (**b**) activity on *C. elegans*. Results are represented as mean ± SEM. Differences compared to the control group were considered significant at *p* ≤ 0.05 (*) and *p* < 0.01 (**).

**Figure 8 foods-13-01040-f008:**
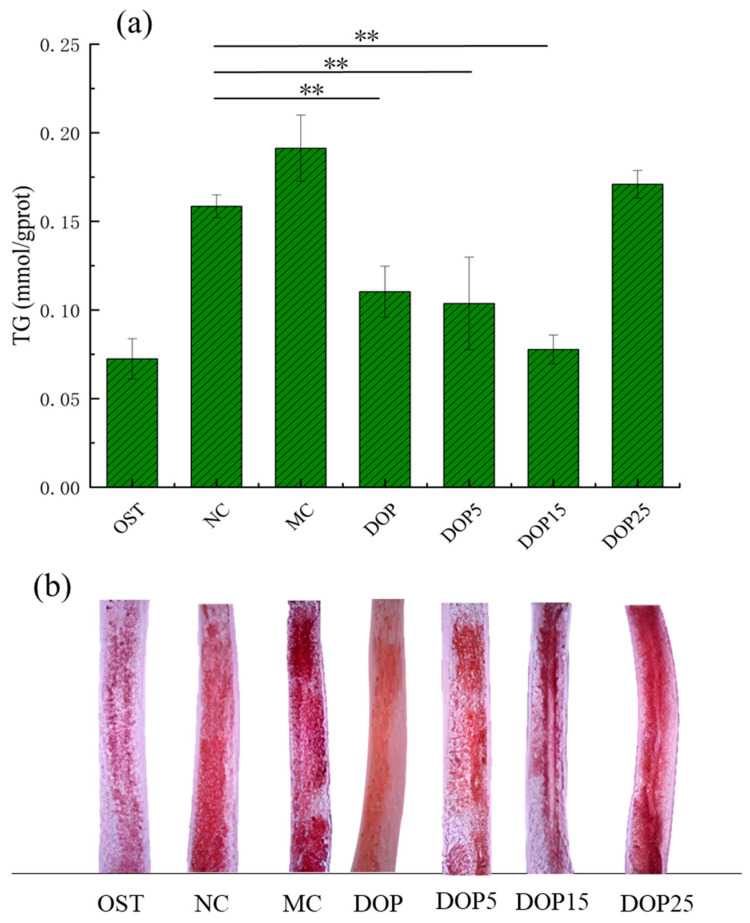
(**a**) TG content of worms in the model group and treatment group. (**b**) A representative picture of ORO staining of worms in the model group and treatment group. Differences were considered significant at *p* < 0.01 (**).

**Figure 9 foods-13-01040-f009:**
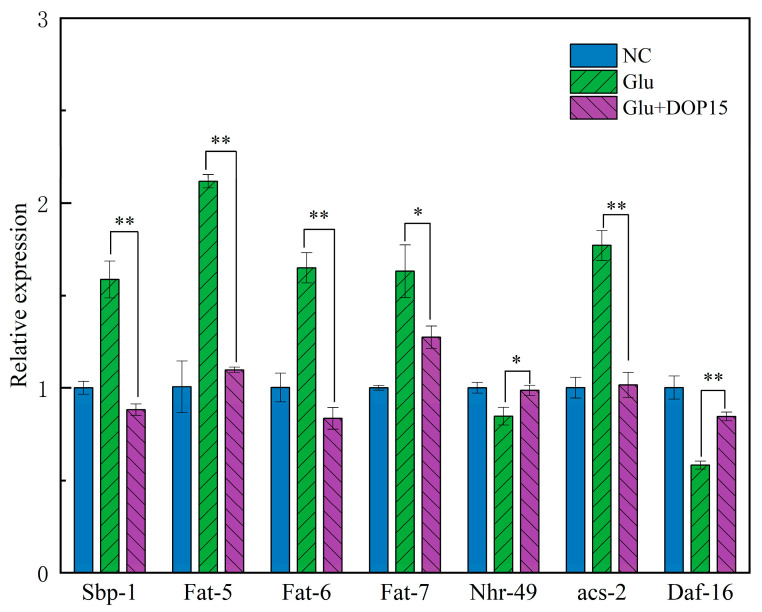
Effect of DOP15 on the lipid metabolism relative gene expression of *C. elegans*. Differences compared to the control group were considered significant at *p* ≤ 0.05 (*) and *p* < 0.01 (**).

**Table 1 foods-13-01040-t001:** Primer list of the genes.

Gene	Direction	Primer (5′-3′)
Sbp-1	F	TGGCGTTCCAATCTATTCGC
	R	CTTGTGGGTTGGCTCCGTT
Fat-5	F	TCGGAGAAGGAGGTCACAAC
	R	TCCCGTTCAGTTTCACAGCC
Fat-6	F	GCGCTGCTCACTATTTCGGA
	R	TGGAAGTTGTGACCTCCCTC
Fat-7	F	TGTTTCACACTTCACGCCAC
	R	CCTCCTTCACCAACGGCTAC
Nhr-49	F	ATCAGATGCCAGATGACGCA
	R	TGCTGTAAAGAGACCGGAGC
acs-2	F	GCGGAGCACATTAAGAAGGC
	R	TGACAGTTCCGAGACCCAAC
Daf-16	F	TACCGGGTGCCTATGGAAAC
	R	AGAGCCGATGAAGAAGCGAC

**Table 2 foods-13-01040-t002:** Effects of DOPs on the average and maximum life of *C. elegans*.

Groups	Average Lifespan (d)	Increase (%)	Maximum Lifespan (d)
MC	17.38 ± 0.85 ^#^	-	28.47 ± 1.01 ^#^
NC	19.12 ± 0.55	-	30.61 ± 1.22
DOP	21.10 ± 0.50 **	10.32	30.00 ± 0.77
DOP5	23.66 ± 1.36 **	23.73	34.65 ± 0.92 **
DOP15	22.02 ± 0.58 **	15.19	33.90 ± 0.73 **
DOP25	20.09 ± 0.25 **	5.06	31.36 ± 1.02

Note: Compared with the negative control group: #, *p* < 0.05 indicates a significant difference. Compared with the model group: **, *p* < 0.01 indicates an extremely significant difference.

## Data Availability

The original contributions presented in the study are included in the article; further inquiries can be directed to the corresponding author.

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
