# Peer review of "Impact of Molecular Weight Variations in Dendrobium officinale Polysaccharides on Antioxidant Activity and Anti-Obesity in Caenorhabditis elegans"

_foods, 2024, doi:10.3390/foods13071040_

Round 1

Reviewer 1 Report

Comments and Suggestions for Authors

This work discusses the optimum molecular weight of polysaccharides derived from Dendrobium officinale for achieving anti-oxidant and anti-obesity effects in C.elegans. The manuscript is suitable for the journal readership and it reads well.

I have the following comments to the authors:

1.     In lines 36-37, the sentence: “Amidst the escalating global obesity epidemic, which now ensnares over a quarter of the world's population with its myriad health complications”

This sentence is incomplete and is difficult to read. It should be restructured.

2.     Referring to lines 63-65, the sentence: “In addition, Xu noted that polysaccharides with a medium molecular weight from camellia seed cake, in contrast to those with higher molecular weights, showed excellent antioxidant properties[19].”

Add a brief explanation of the reason for the observed behaviour.

3.     In line 101, provide the complete term, for DPTA.

4.     In lines 222 and 223, please provide adequate references here for the followed protocols for Oil red O staining and quantification of triglycerides.

5.     In line 271, what does "VC" stand for? Please provide the complete term.

6.     In line 328, Table 1 should be corrected to Table 2.

7.     In line 502, reference [28] the author's name “Sevng” should be corrected to “Sevag”

8.     In the discussions section the authors should bring examples of the applications of polysaccharides derived from Dendrobium officinale in functional foods. Currently, the discussion on their application in foods is missing in the manuscript and considering that the special issue is on nutraceuticals and functional foods, discussing the application of the tested polysaccharides in foods is necessary.

Reviewer 2 Report

Comments and Suggestions for Authors

- 51: please provide a full name of Dendrobium officinale [Dendrobium officinale Kimura et Migo]

-85: standards and reagents: please provide catalog nr of products, like kits etc. 

- 114: Analysis of Polysaccharide Components: Which detector was used UV-ViS or refractive index detector (RID), which is more dedicated to detect sugars.

- 424: please discuss your results with those available in literature.

- what are the limitations of the results obtained and their application regarding human or animals model?
